



# Regional effect on urban atmospheric nucleation

Imre Salma[1], Zoltán Németh[1], Veli-Matti Kerminen[2], Pasi Aalto[2], Tuomo Nieminen[2], Tamás Weidinger[3], Ágnes Molnár[4], Kornélia Imre[4], Markku Kulmala[2]

[1]Institute of Chemistry, Eötvös University, H-1518 Budapest, P.O. Box 32, Hungary
[2]Department of Physics, FI-00014 University of Helsinki, P.O. Box 64, Finland
[3]Department of Meteorology, Eötvös University, H-1518 Budapest, P.O. Box 32, Hungary
[4]MTA-PE Air Chemistry Research Group, H-8201 Veszprém, P.O. Box 158, Hungary

*Correspondence to*: I. Salma (salma@chem.elte.hu)

**Abstract.** Secondary aerosol particle production via new particle formation (NPF) has been shown to be a major source for global aerosol load. It has been also observed frequently in urban environments affecting the human health. Here, we investigate the effect of regional NPF on urban aerosol load under well-defined atmospheric conditions. The Carpathian Basin, the largest orogenic basin in Europe, represents an excellent opportunity for exploring these interactions. Based on long-term observations, we revealed that NPF seen in a central large city of the basin (Budapest) and its regional background occur in a consistent and spatially coherent way as result of a joint atmospheric phenomenon taking place over large horizontal scales. We found that NPF events at the urban site are usually delayed by >1 hour relative to the rural site or even inhibited above a critical condensational sink level. The urban processes require higher formation rates and growth rates to be realised, by mean factors of 2 and 1.6, respectively, than the regional events. Regional- and urban-type NPF events sometimes occur jointly with multiple onsets, while they often exhibit dynamic and timing properties which are different for these NPF types.

## 1 Introduction

New aerosol particle formation (NPF) and consecutive particle growth processes in the atmosphere (Kulmala et al., 2013) were first identified in clean environments (Weber et al., 1995; Mäkelä et al., 1997), and the NPF occurrence frequency and its contribution to particle number concentrations were later found to be substantial in the global troposphere (Kulmala et al., 2004; Spracklen et al., 2006). Particles originating from these processes affect the Earth's radiation balance mainly by acting as cloud condensation nuclei (CCN, Kerminen et al., 2012; Carslaw et al., 2013), and their contribution to the total number of CCN can be up to 50% or even more (Merikanto et al, 2009). Recently, NPF has been proved to be common in polluted environments including large cities (Woo et al., 2001; Baltensperger et al., 2002; Alam, et al., 2003; Wehner et al., 2004; Salma et al., 2011; Dall'Osto et al., 2013; Xiao et al., 2015) as well, but the connections between the urban and regional air have remained unknown. Despite the fact that NPF and consecutive growth can also interact with urban climate and can contribute to the public's excess health risk from nanoparticle exposure (Salma et al., 2014, 2015).

The Carpathian Basin offers excellent conditions for such studies with its extension, moderate climate and topographically discrete character, and because it contains a large city, Budapest, at its central part. We characterized atmospheric conditions of the city centre and near-city background of Budapest by performing continuous measurements of aerosol, air pollutant gases and meteorological data for two 1-year long time intervals. We completed similar measurements in the rural background at K-puszta station representing the regional atmospheric conditions over the same time intervals. The major objectives of the present paper are to identify and quantify both important similarities and differences between the urban and regional NPF types, and to investigate the interactions and associations between urban air in Budapest and regional rural atmosphere in the background of the city with respect to NPF.





## 2 Methods

### 2.1 Measurement locations and periods

The Carpathian Basin (also known as the Pannonian Basin) is a topographically discrete unit in the south-eastern part of Central Europe surrounded by the Eastern Alps, the Carpathian Mountains, the Balkan Mountains and the Dinaric Alps (Karátson, 2006; Fig. 1)). Its typical dimensions in the N–S and W–E directions are approximately 450 and 650 km, respectively with a territory of approximately 330 thousand $km^2$. Three climate zones, namely maritime temperate (typical for the mainland), warm summer continental (typical for the eastern-central part) and Mediterranean (typical for the southern part) climate zones meet in the region. Annual insolation is 4200–4700 MJ $m^{-2}$ with a maximum in the central part (Spinoni et al., 2014). Annual sunshine duration is between 1850 and 2200 h, which results in an annual relative sunshine duration of 40–47% with a minimum of 15–20% in December, and a maximum of 55–65% in August. Annual mean air temperature at sea level is approximately 10 °C, which is larger by about 2.5 °C than the global annual mean temperature for corresponding latitudes. There are differences up to 3–4 °C in the annual mean temperatures among various sub-basins. Mean temperature range is substantial; its annual mean is between 21 and 26 °C, and has an increasing tendency to the E as the influence of the continental climate zone. Annual mean rainfall in the lowland is between 600 and 700 mm with a minimum in the central part of the territory. The Carpathian Basin is located in the belt of the Prevailing Westerly Winds. The bounding mountains, however, represent important barriers to the global wind pattern. The wind directions and speeds within the basin are substantially modified by passing weather systems, inner mountains and local radiation gradients. As a result, the surface westerly winds arriving to the basin are usually spread to N winds prevailing in the western part, and to NW winds occurring especially in the upper central part of the basin. In the southern part of the Great Hungarian Plain, southerly winds often occur, while in the eastern part of the basin, the prevailing wind directions are NE or N particularly in winter. Annual mean wind speed at a height of 10 m is between 2 and 4 m $s^{-1}$ with smaller values in autumn and larger values in spring. Weather situations within the basin are generally uniform. For some limited time intervals, rather distinct weather conditions can be realised in different territories of the basin due to the geographical, meteorological and climate properties and features mentioned above. The land of the basin is mostly used for intensive agriculture and farming, and larger forested areas with deciduous, coniferous or mixed wood occur in the inner and bounding mountains. Total number of inhabitants in the basin is estimated to 26 million, and its largest city is Budapest with 2.5 million inhabitants in the metropolitan area.

The data evaluated in the present study were obtained in Budapest and at K-puszta measurement station (Fig. 1). The geographical area between the two sites is plain. The measurements in Budapest were carried out in the central part of the city (Lágymányos Campus of the Eötvös University, 47.474° N, 19.062° E, 114 m above mean sea level, asl) in an open area near the river Danube continuously from 3 November 2008 to 2 November 2009 (Salma et al., 2011). The location represents well-mixed urban air in the city centre. Another 1-year long continuous measurement campaign was realised at the NW border of Budapest in a wooded area (Konkoly Astronomical Observatory of the Hungarian Academy of Sciences, 47.500° N, 18.963° E, 478 m asl) from 19 January 2012 to 18 January 2013. This site represents the near-city background. The K-puszta measurement station (46.967° N, 19.586° E, 127 m asl) is situated on the Great Hungarian Plain in a distance of 71 km SE from Budapest. The station is located on a forest clearing with its near-scale surrounding dominated by mixed forest (62% coniferous, 28% deciduous) and grassland (10%). The station represents the rural background. It is involved in the European Monitoring and Evaluation Programme (EMEP station no. HU0002R). Two 1-year long data sets which correspond exactly to those considered for Budapest were selected for the evaluations.



## 2.2 Measurements

The principal measuring system was a flow-switching type differential mobility particle sizer (DMPS) at both sites. Its main components are radioactive bipolar charger, Nafion semi-permeable membrane dryer, differential mobility analyser and a butanol-based condensation particle counter (TSI CPC3775 in Budapest and CPC3772 at K-puszta station, Aalto et al., 2001).

Particles with an electrical mobility diameter from 6 to 1000 nm (Budapest), and from 10 to 800 nm (K-puszta) are recorded in their dry state in 30 channels. Sample flow was 2.0 L min$^{-1}$ (Budapest) or 1.0 L min$^{-1}$ (K-puszta) in high-flow mode, and 0.3 L min$^{-1}$ in low-flow mode. Time resolution of the measurement was approximately 10 min. There was no upper size cut-off inlet applied to the sampling lines, and a weather shield and insect net were only adopted. The DMPS measurements were performed according to the recommendations of the international technical standard (Wiedensohler et al., 2012). Synoptic

meteorological data were obtained from regular measurement stations of the Hungarian Meteorological Service operated in Budapest (no. 12843) and at the military airport in Kecskemét (no. 12970). Standardised meteorological measurements of air temperature ($T$), relative humidity (RH), wind speed (WS) and wind direction (WD), and cloudiness ($n$) are recorded at these stations with a time resolution of 1 h. Global radiation (GRad) was calculated from the measured meteorological data by method of Holtslag and Van Ulden (Foken, 2006; Weidinger et al., 2008). Planetary boundary layer (PBL) height was obtained

from the ECMWF Integral Forecast System based on the ERA-Interim reanalysis with a spatial resolution of 0.5°×0.5° and time resolution of 3 h (Dee et al., 2011). Concentration of atmospheric criteria pollutants were obtained from the closest measurement station of the National Air Quality Network in Budapest, and in Százhalombatta for K-puszta station both located in the upwind prevailing direction from the measurement sites. Ozone and PM$_{10}$ mass were recorded directly at the K-puszta station. Standardised measurements of SO$_2$ (by UV fluorescence, Ysselbach 43C), PM$_{10}$ mass (by beta-ray absorption, Thermo

FH62-I-R) and O$_3$ (by UV absorption, Ysselbach 49C) with a time resolution of 1 h are performed at the stations. The DMPS data in Budapest and at K-puszta station were available for more than 90% and 70%, respectively of the total number of days, while coverage of the meteorological and pollutant data were >80%.

## 2.3 Evaluation

Overall treatment of the measured DMPS data was performed according to the procedure protocol recommended by Kulmala

et al. (2012). The individual size distributions were fitted by lognormal functions using the DoFit algorithm (Hussein et al., 2004). Identification of a NPF and subsequent particle growth process was accomplished by using the algorithm of Dal Maso et al. (2005). Formation rate of particles ($J_d$) with a diameter $d$, and condensation sink (CS) were computed according to Kulmala et al. (2001) and Dal Maso et al. (2002). Growth rate (GR) of particles for a size interval of 6–25 nm was determined by log-normal distribution function method (Kulmala et al., 2012). The earliest estimated time of the beginning of a nucleation

($t_1$), the latest estimated time of the beginning of a nucleation ($t_2$), and the ending time of the particle growth process ($t_e$) were derived by a comparative method (Németh and Salma, 2014). The time $t_1$ also includes the time shift that accounts for the particle growth from the stable neutral cluster mode at (1.5±0.4) nm (Kulmala et al., 2007) to the smallest detectable diameter limit of the DMPS systems. Gas-phase H$_2$SO$_4$ proxy value was derived as [SO$_2$]×GRad/CS for intensities >10 W m$^{-2}$, and the scaling factor $k$ between the proxy value and H$_2$SO$_4$ concentration was estimated by an empirical relationship of $k$=1.4×10$^{-7}$×

GRad$^{-0.70}$ (Petäjä et al., 2009).

Retrospective movement of the air masses was assessed by backward trajectories, which were generated by using the air parcel trajectory model HYSPLIT v4.9 with an option of vertical velocity mode (Draxler and Rolph, 2013). Embedded Global Data Assimilation System meteorological database was utilised for the modelling, which yields the calculated results on a 1-degree

latitude-longitude grid. Trajectories arriving to the receptor sites at a height of 200, 500, 2300 m above ground level were calculated. For the NPF event days, the start time of the backward modelling was set at the ending time of the particle growth



($t_e$), and the end time of the computer run was fixed at the earliest beginning of the nucleation ($t_1$). For the non-event days, the end and start times of the modelling were set at 13:00 and 1:00 UTC+1, respectively.

Correlation analysis between the joint two-year long data sets for Budapest and K-puszta station was performed on a daily
basis. Index of occurrence 1 was assigned to NPF event days, 0 to undefined and missing days, and −1 to non-event days. Pearson correlation coefficient and its transformed $t$-value was calculated as:

$$t = r \sqrt{\frac{n-2}{1-r^2}} \, , \tag{1}$$

for $n$=731 items and assuming Student's $t$-distribution with $n$–2 degree of freedom.

Wind speed data measured at a height of 10 m above the ground were recalculated to a height of 200 m by using the power law approach with an exponent of 0.2 (Irwin, 1967). These WS data were averaged for the area between Budapest and K-puszta station considering different time intervals. The following 4 cases were considered: NPF events identified at both measurement sites (BpY&KpY), event in Budapest and no event at K-puszta station (BpY&KpN), no event in Budapest and event at K-puszta station (BpN&KpY), and no event in both Budapest and K-puszta station (BpN&KpN). For the case
BpY&KpY, the WS data for the site with the earlier NPF event were selected from the time $t_1$−1 h of the earlier site to the time $t_1$ of the site with the delayed NPF event, while the WS data for the site with the delayed event were selected from the time $t_1$ of the earlier site to the time $t_1$+1 h of the delayed site. Finally, the two selected data sets were averaged to a synoptic WS value. For the cases BpY&KpN and BpN&KpY, the mean WS value was obtained by averaging the two WS data sets between the times $t_1$ and $t_e$. For the case BpN&KpN, the mean WS value was derived by averaging jointly both WS data sets from the
mean time $t_1$ to the mean time $t_e$. These selections represent sensible and representative approximations to reality. As the next step, comparison of the delay time with the travel time of the synoptic wind was expressed by:

$$\tau = \frac{\Delta t_1}{D} \, WS \, , \tag{2}$$

where $D$=71 km is the distance between Budapest centre and K-puszta station. If the ratio $\tau \rightarrow 1$ then the nucleating air mass was likely transported by advection from the upwind site to the downwind site. If $\tau$ is substantially smaller than 1 then the NPF
event at the downwind site could not be transferred by advection from the upwind site but it was rather formed somewhere else, while $\tau$ >1 is often caused by large (>7 m s$^{-1}$) WSs.

## 3 Results and discussion

### 3.1 Similarities in NPF occurrence

The annual mean frequencies of NPF events were substantial (Kulmala et al., 2004, and references therein; Dal Maso et al.,
2005; Manninen et al., 2010; Borsós et al., 2012; Dall'Osto et al., 2013) at our measurement sites (Table 1). This indicates that the typical meteorological and chemical conditions (e.g., higher annual mean $T$ inside the basin, presence of forested areas and availability of SO$_2$ precursor) within the Carpathian Basin favour the occurrence of NPF. The seasonal variability of the NPF frequency was very similar at both sites with a minimum in winter and two local maxima, one in spring and the other in autumn (Fig. 2). The spring maximum took place in April during the first year and in March during the second year.
Such a shift can be caused by inter-annual differences in meteorological conditions (Dal Maso et al., 2005; Hamed et al., 2010) or biogenic cycling through emissions of volatile organic compounds (VOCs) from vegetation (Riipinen et al., 2011; Riccobono et al., 2014). The two measurement sites seemed to respond identically to these influences.





In order to investigate more thoroughly whether and how the occurrence of NPF was connected between the two measurement sites, and over the whole basin, we made a statistical analysis and investigated air mass transport effects. For the occurrence of NPF, the joint two-year long data set resulted in a correlation coefficient of 0.429 with Student's $t$-value of 12.829. The critical Student's $t$-value at a statistical confidence level of 99.99% is 3.912, which means that there was a significant linear

relationship in the occurrence of the NPF between the 2 sites. In most cases, the particle growth associated with NPF (the presence of the banana curve) could be traced for 8–13 h, during which time the air parcel containing nucleated particles travelled a distance comparable to the dimensions of the basin. This result is consistent with other modelling studies on the spatial extent of NPF (Crippa and Pryor, 2013; Németh and Salma, 2014; Pietikäinen et al., 2014).

### 3.2 Incidence of NPF events

The geographical straight line between the two measurement locations is identical with the prevailing wind direction (NW) in the area, which makes it possible to compare the exact timing of the NPF processes at these sites to the air mass advection between the sites (Table 2). Air mass backward trajectories derived for all days were grouped into the following cases: trajectories in parallel with the geographical connecting line arriving from NW, in the parallel direction with the connecting line arriving from SE, trajectories in largely perpendicular directions to the connecting line, and trajectories in unclassified

directions. Days with quantifiable NPF event and non-event days in the two-year long data set were considered in this evaluation. For simultaneous days (BpY&KpY), the NPF events in Budapest occurred later than at K-puszta station in 75% of the relevant days with a mean delay time of 1 h 54 min. There were 4 days when the NPF events in Budapest happened earlier than at K-puszta station but their mean delay time involved a relative uncertainty of >80%. For the parallel direction from SE, the NPF events in Budapest occurred later than at K-puszta station in 85% of the relevant days with a mean delay time of 1 h

11 min. The opposite timing (on 3 days only) yielded a mean delay time with a rather high uncertainty. For the parallel direction from NW, the NPF events in Budapest occurred later than at K-puszta station in 33% of the relevant days with a mean delay time of 1 h. This was promoted by the air mass transport. During the days when NPF was taking place at both sites (about 10% of the days), the mean and standard deviation of the ratio $\tau$ (Eq. 2) were 0.34±0.25. This indicates that the air masses only reached approximately one third of the geographical distance when the NPF had already started at the downwind site. Hence,

the NPF observed at the downwind site was not because of air mass advection from the upwind site, but rather took place simultaneously over large distances in the basin. At present knowledge, advection of nucleating air masses cannot be excluded only in a few cases.

The cumulative results outlined in Sects. 3.1 and 3.2 show that the NPF events observed in the city and its rural background

appear in a consistent and spatially coherent way. We interpret this as the result of a common atmospheric nucleation phenomenon in the region with some urban influence.

### 3.2 Site-to-site variability of NPF properties

There were, however, clear differences in the NPF process between the city and rural background. The annual mean NPF occurrence frequencies of 24% and 28% for Budapest sites were smaller by factors of 1.3 and 1.4, respectively, than for K-

puszta station (Fig. 2). Moreover, a continuously supressing tendency from the rural background through the near-city background to the city centre was found. The mean values of $J$ and GR for central Budapest exceeded that for K-puszta station by factors of 2 and 1.6, respectively, when considering the whole measurement data set (Table 3). This is consistent with the idea that particles capable of escaping coagulation scavenging need to grow faster in polluted air compared to cleaner environments. Larger GR values are typical for polluted urban atmospheres, e.g., for New Delhi: GR=11.6–167 nm h$^{-1}$

(Kulmala et al., 2005), for Mexico City: 15–40 nm h$^{-1}$ (Iida et al., 2008), for South Africa: 3–21 nm h$^{-1}$ (Laakso et al., 2008) or for Shanghai: 11.4±9.7 nm h$^{-1}$ (Xiao et al., 2015). Quantifiable-NPF events at K-puszta station started from 5:35 (on 28–





04–2009) to 12:46 (on 15–06–2009). The starting time for the near-city background and city centre of Budapest ranged from 6:55 (on 25–08–2012) to 12:24 (on 11–10–2012), and from 6:11 (on 28–04–2009) to 11:43 (on 09–05–2009), respectively.

The relationship between the major source and sink for gas-phase $H_2SO_4$ for the event days and non-event days is shown in Fig. 3 separately for Budapest and K-puszta. The data for nucleation days are means averaged from the time $t_1$ to time $t_2$ of a NPF event. The data for non-event days are means derived by averaging between the overall mean times $t_1$ and $t_2$. All time data were expressed in UTC+1. Figure 3 suggests that the CS effectively suppresses the NPF at values larger than about $20 \times 10^{-3}$ $s^{-1}$ at both sites. The common limiting CS value is likely related to some environmental features in the Carpathian Basin. At smaller values of CS, a large number of non-event days is located above the dividing line in Fig. 3, which can be explained by factors other than CS (e.g., high concentrations of inhibiting chemical species (Kiendler-Scharr et al., 2009), low concentrations of stabilizing compounds such as oxidised VOCs (Riccobono et al., 2014), $NH_3$ and amines (Almeida et al., 2013), or large RH (Hamed et al., 2011) being active in suppressing the NPF. The much lower fraction of event days locating below the same dividing line supports the principal role of $H_2SO_4$ in the NPF process (Sipilä et al., 2010), especially for Budapest. The median values of the $H_2SO_4$ proxy concentration for the event days in Budapest ($4.8 \times 10^6$ molecules $cm^{-3}$) and K-puszta ($4.7 \times 10^6$ molecules $cm^{-3}$) were larger by factors of 1.4 and 1.2, respectively, than for the non-event days at these two sites (Table 4). The mean diurnal variation of the $H_2SO_4$ proxy was also very different between the event and non-event days, as well as between the two sites (Fig. 4). At the mean NPF starting time for Budapest of $t_1$=9:25 UTC+1, the $H_2SO_4$ proxy appears to be separated into larger values ($>45 \times 10^4$ µg $m^{-5}$ W s $\propto 6 \times 10^6$ molecules $cm^{-3}$) for event days and smaller values ($<25 \times 10^4$ µg $m^{-5}$ W s $\propto 3 \times 10^6$ molecules $cm^{-3}$) for non-event days. This partitioning is related to the availability of $H_2SO_4$ and seems to explain the observed variability in the occurrence of NPF in Budapest. In contrast, the $H_2SO_4$ proxy for K-puszta station at the mean NPF starting time of $t_1$=8:40 UTC+1 lie in a narrow band, except for the case BpY&KpY. The $H_2SO_4$ proxy utilised in the present study takes only into account the main atmospheric oxidation of $SO_2$ by the OH radical. Recent field observations supported by laboratory experiments and theoretical considerations point to capacities of stabilized Criegee intermediates (sCIs) in forests to oxidize $SO_2$ into $H_2SO_4$, with a contribution of up to 33–46% of $H_2SO_4$ concentration at ground level (Mauldin III et al., 2012; Boy et al., 2013). Stabilized CIs are formed by ozonolysis of unsaturated organics, including terpenoid compounds which are emitted in large amounts by plants. This biogenically-related mechanism cannot be excluded for the NPF events observed just after the sunrise at K-puszta station.

Based on the $H_2SO_4$ proxy concentration, the mean contribution of $H_2SO_4$ condensation to the particle GR was estimated to be 12.3% and 11.8% for Budapest and K-puszta, respectively. This indicates that other chemical species, presumably organic compounds, have a large influence on the growth of newly-formed particles in the basin, and possibly also on the NPF process itself. The increase in the GR with particle size reported earlier for the K-puszta station (Yli-Juuti et al., 2009) is strongly in line with this view. Previous studies have identified extremely low-volatile organic compounds (ELVOCs) playing an important role in both the NPF and particle growth. Such compounds were first predicted by Kulmala et al. (1998) and later identified from α-pinene ozonolysis in smog chamber experiments (Ehn et al., 2014). ELVOCs appear to be formed with substantial mass yields under atmospherically relevant conditions and their dimers seem large enough to act as nano-condensation nuclei for their further irreversible growth (Ehn et al., 2014). Our results indicate clear contribution of compounds other than $H_2SO_4$ in the initial and subsequent growth processes. Since the saturation vapour pressure of those compounds need to be very low, it is probable that ELVOCs are the main contributors for this in the Carpathian Basin.

### 3.3 Spatial variability of NPF occurrence on sub-regional scale

In order to find out the primary causes for the sub-regional differences, we derived median values of relevant meteorological data and air pollutant concentrations separately for the following cases: BpY&KpY, BpY&KpN, BpN&KpY and BpN&KpN




(Table 4). Since the details of the possible oxidation of $SO_2$ by sCIs are not exactly known, median $O_3$ concentrations for the previous day of a NPF event were also added for the K-puszta station. The averaging was first performed from time $t_1$ to time $t_e$. In the case of BpY&KpY, the actual time parameters for each measurement site were utilized. In the cases of BpY&KpN and BpN&KpY, the time parameters of the NPF event for one of the measurement sites were used for the other, non-event site

as well. In the case of BpN&KpN, overall mean time parameters $t_1$ and $t_e$ were adopted as averaging limits. For GRad and $H_2SO_4$ concentration, the ending time $t_e$ was replaced by time $t_2$. These averaging limits represent time intervals in which the NPF event and particle growth are the most pronounced. Finally, the mean values for particular days were further averaged separately for the four combination cases mentioned above.

Variability for GRad, RH, $T$ and PBL height are strongly biased by the seasonal cycle of solar radiation via the distribution of monthly NPF frequency, and, therefore, their tendencies are to be approached with a special care. Variables $PM_{10}$ and WS did not seem to contribute substantially to the explanation of the heterogeneity, while $N_{10-100}$ just reflected that NPF increases the number concentration of pre-existing particles extensively (by a factor of 2–3 for Budapest, Salma et al., 2011). Mean concentrations of $O_3$ were larger at K-puszta station than in Budapest, and it was also larger on NPF days than on non-event

days, which indicates larger photochemical activity in the former case. Cloudiness showed some weak reciprocal tendency with NPF. Occurrence of NPF events did not seem to be sensitive to $SO_2$ concentration, which suggests that this precursor gas was available in excess for most of the time. In spite of the fact that the estimated reduction of $SO_2$ emission between 1990 and 2004 was more than 60% in most European countries, including Hungary (Hamed et al., 2010). Condensation sink in Budapest exhibited a general dependency that NPF events occur preferably on days with low CS values. It was larger for the

non-event days than for the event days by approximately 50%. This implies that the CS affected the NPF in the Budapest area, and that it can have preventing influence on the events. In contrast, the mean CS values for K-puszta station showed much less or even little effect.

For the mixed cases of BpN&KpY and BpY&KpN, the difference between the two data was considered to be significant if

they diverged at least by a factor of 2. For cloudiness, a difference larger than 2 okta at an absolute value larger than 5 okta was required, similarly to RH, for which a difference of 15% at an absolute values of >70% were needed for a significant difference (Boy and Kulmala, 2002). These criteria are in line with understanding of the conditions for NPF process (Hamed et al., 2011), and represent sensible approximations to reality. Of 58 cases of BpN&KpY, on 19 days, the CS was significantly larger, on 8 days, the $H_2SO_4$ proxy was significantly smaller, on 8 days, both the CS was significantly larger and proxy was

significantly smaller, on 4 days, the RH was significantly larger and large, on 3 days, the cloudiness was significantly larger and large, and on 1 day, both the proxy was significantly smaller and the cloudiness was significantly larger and large in Budapest than at K-puszta station. These explain 77% of the investigated days. On further 2 days, some important data were missing, and, therefore, they evaluation was hindered. In 23% of the relevant days, we could not prove plausible causes for the spatial difference. About 15% of these days were realised in summer, late spring or early autumn, when the $O_3$ production

and its chemical reactions could be important, and there was an indication of increased $O_3$ concentration overnight before the NPFs. It is hypothesized that most of these unexplained events at K-puszta station were explicit cases for sCI oxidation mechanism. For the case BpY&KpN, 11 days (67% of the days) could be explained by larger cloudiness, RH, CS and WS and/or lower $SO_2$ at K-puszta station than in Budapest. The number of days with missing data was 3. Possible factors for the unexplained days can also include different weather systems at or weather front between the measurement sites, different air

masses or the conservative selection criteria applied. The area between Budapest and K-puszta station is inhabited, and the effects of settlements on the atmospheric environment also limit the homogeneity of larger air masses. By analyzing the data set, we can conclude that the two major candidate for explaining the differences in the occurrence of NPF are the higher CS in Budapest and smaller gas-phase $H_2SO_4$ concentration at K-puszta.





### 3.4 Regional- and urban-type NPF events

We observed NPF and subsequent particle growth events with at least two consecutive onsets on some days, in particular in the near-city background. There were 8 NPF events with double start of 43 quantifiable events there. A surface plot displaying a typical double start together with temporal evolution of some related quantities is shown in Fig. 5 as example. The quantities

which are related to the NPF events varied substantially and rapidly in time, while the concentration $N_{100-1000}$, which represents a larger region, stayed stable. This means that there was no extraordinary change in the dynamics of the PBL or weather situation during the time interval of the two onsets. In addition, there was no indication of sudden change in the local WD, WS and GRad data, or in the air mass origin and path during the relevant time intervals. Based on these arguments, interrupted and renewed (started over) NPF and particle growth process due to the changes in local meteorology, or to two different air masses

transported to the measurement site can be largely excluded, and there must be other primary reasons for the double starts.

The characteristics of NPF for the event consisting of two onsets were quite different from each other (Table 5). The mean ratio and standard deviation of $J_6$ and GR between the two onsets were 2.5±1.0, and 1.8±0.5, respectively, and the mean difference and standard deviation between the corresponding starting times were (2:12±0:36) h. The individual $J_6$s and GRs

separately for the NPF events with single start, and for the two onsets of the NPF events with double start are shown in Fig. 6. The dynamic properties of the onset 2 (later event) were always larger than those of the corresponding onset 1 (sooner event). In addition, the mean NPF characteristics of the sooner onsets were similar to those of the single-onset events within the uncertainty interval, indicative of their common cause, and also similar to those for the rural background data (Yli-Juuti et al., 2009). At the same time, the mean NPF characteristics of the later onsets were close to those for the city centre data (Salma et

al., 2011). This suggests that the later events occurred in a more polluted air, which was unambiguously of urban origin.

The interpretation is also supported by the tendency that the later onsets generally happened when the $H_2SO_4$ proxy was high, and they might also be associated with larger $J$s and GRs (Fig. 8). At K-puszta station, there was no NPF observed on 2 of the 8 related days, and for 1 day, some important data were missing, and, therefore, the number of available data points for K-

puszta station is limited. It appears, however, that the overall conditions for the NPF process are met for the whole region. If this is realised then a regional event likely occurs, which can be accompanied by an urban-type NPF at a later time. This may take place, for instance, by mixing regional and urban air parcels that exhibit different properties, which is mainly governed by local PBL dynamics and urban heat island effects. We relate these distinctions to urban influence. Our interpretation is supported by a previous observation of NPF event with multiple onsets in semi-clean savannah and industrial environments

(Hirsikko et al., 2013), and fits very well into the existing ideas on mixing regional and urban air parcels that exhibit different properties such as precursor concentrations, $T$ and RH, and which is mainly governed by local PBL dynamics (Nilsson and Kulmala, 1998).

### 4 Conclusions

The cumulative evidence on urban/non-urban similarities and differences in NPF and growth processes suggests that NPF

events observed in a well-defined and large enough atmospheric environment (such as the Carpathian Basin) appear in a consistent and spatially coherent way as result of a common atmospheric phenomenon taking place over large horizontal scales. The conclusion does not necessary mean that NPF events happen uniformly either in time or space within the whole basin since subtle differences in the spatial concentration gradients, PBL dynamics, or local quenching effects for supersaturation over some territories can cause systematic or accidental spatial and temporal variability and alterations. In this way, regional

NPF events also interact and, hence, are modified, extended or transformed by urban influences and properties as well. There





are strong relationships between cities and their regional environments in that sense, and the joint regional- and adherent urban-type NPF events can influence the urban climate, and can contribute to the excess health risk from nanoparticle exposure of the public. This later observation opens new potentials and challenges for further health related studies on ambient aerosol nanoparticles as well.

**Author contributions.** I.S. and M.K. designed the study. Z.N. and P.A. conducted the data collection and treatment. Z.N., T.N., I.S., T.W., Á.M., and K.I. performed the data analyses and modelling calculations. I.S., V.-M.K., M.K. and Z.N. were involved in the interpretation of the results. I.S. and V.-M.K. wrote the manuscript. M.K. improved the content of the manuscript. The authors declare no competing financial interests.

**Acknowledgements.** The research was financially supported by the Hungarian Scientific Research Fund (no. K84091), ERC-Advanced grant ATMNUCLE (no. 227463) and Academy of Finland (Center of Excellence project, no. 1118615). The authors thank A. Kern and Zs. Ungvári for preparation of Fig. 1, and E. Keszei (all of the Eötvös University) for discussions on statistical methods. The PBL data were obtained from the courtesy of I. Ihász of the Hungarian Meteorological Service.

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



**Table 1.** Number of days with obvious NPF event, unambiguously no NPF event, undefined days, and missing days in Budapest and at K-puszta station for two 1-year long time intervals in 2008–2009 and 2012–2013.

| Time interval/class | Budapest | K-puszta |
|---|---|---|
| 2008–2009 (365 days) | | |
|    Events | 83 | 100 |
|    Non-events | 229 | 146 |
|    Undefined | 34 | 78 |
|    Missing | 19 | 41 |
| 2012–2013 (366 days) | | |
|    Events | 96 | 125 |
|    Non-events | 231 | 180 |
|    Undefined | 19 | 40 |
|    Missing | 20 | 21 |



**Table 2.** Number and relative frequency of air mass trajectories arriving to Budapest and K-puszta station in parallel direction from NW and SE, in perpendicular directions, and in unclassified (other) directions relative to the Budapest–K-puszta station geographical connecting line for the conditions that there were NPF events identified at both sites (BpY&KpY), event in Budapest and no event at K-puszta station (BpY&KpN), no event in Budapest and event at K-puszta station (BpN&KpY), and no event at both sites (BpN&KpN).

| Trajectory direction | BpY&KpY | | BpY&KpN | | BpN&KpY | | BpN&KpN | |
|---|---|---|---|---|---|---|---|---|
| | [#] | [%] | [#] | [%] | [#] | [%] | [#] | [%] |
| Parallel from NW | 23 | 36 | 5 | 33 | 9 | 16 | 91 | 34 |
| Parallel from SE | 20 | 31 | 2 | 13 | 28 | 48 | 39 | 14 |
| Perpendicular | 16 | 25 | 8 | 54 | 21 | 36 | 136 | 50 |
| Other | 5 | 8 | 0 | 0.0 | 0 | 0.0 | 4 | 2 |
| All | 64 | 100 | 15 | 100 | 58 | 100 | 270 | 100 |



**Table 3.** Mean $J_6$, GR and starting time parameter $t_1$ with standard deviations for quantifiable regional NPF events in Budapest and at K-puszta station separately for two 1-year long time intervals in 2008–2009 and 2012–2013. Numbers of days with quantifiable NPF events are also shown. The measurements in Budapest were performed in the city centre in 2008–2009, while they were accomplished in the near-city background in 2012–2013.

| Time interval/property | Budapest | K-puszta |
|---|---|---|
| 2008–2009 (365 days) | | |
| # of quantifiable events | 31 | 45 |
| $J_6$ [cm$^{-3}$ s$^{-1}$] | 4.2±2.5 | 1.9±1.5 |
| GR [nm h$^{-1}$] | 7.7±2.4 | 4.8±2.3 |
| $t_1$ [hh:mm UTC+1] | 9:25±1:11 | 8:48±1:33 |
| 2012–2013 (366 days) | | |
| # of quantifiable events | 43 | 55 |
| $J_6$ [cm$^{-3}$ s$^{-1}$] | 2.1±1.5 | 1.8±1.4 |
| GR [nm h$^{-1}$] | 5.1±1.5 | 4.2±2.1 |
| $t_1$ [hh:mm UTC+1] | 8:44±1:10 | 8:31±1:27 |





**Table 4.** Medians of gas-phase $H_2SO_4$ concentration, condensation sink (CS), $SO_2$ concentration, global radiation (GRad), relative humidity (RH), air temperature ($T$), wind speed (WS), cloudiness ($n$), planetary boundary layer (PBL) height, $O_3$ concentrations on the actual day and previous day, aerosol particle number concentration in the diameter range from 10 to 100 nm ($N_{10-100}$), $PM_{10}$ mass concentration, for the time interval when NPF events were identified in both Budapest and K-puszta station (BpY&KpY), event in Budapest and no event at K-puszta station (BpY&KpN), no event in Budapest and event at K-puszta station (BpN&KpY), and no event in both Budapest and K-puszta station (BpN&KpN).

| Variable | BpY&KpY | | BpY&KpN | | BpN&KpY | | BpN&KpN | |
|---|---|---|---|---|---|---|---|---|
| | Budapest | K-puszta | Budapest | K-puszta | Budapest | K-puszta | Budapest | K-puszta |
| $[H_2SO_4]\times10^{-6}$ [molecules cm$^{-3}$] | 5.1 | 5.4 | 4.4 | 4.3 | 3.4 | 4.0 | 3.3 | 3.5 |
| CS$\times10^3$ [s$^{-1}$] | 7.9 | 6.6 | 8.8 | 6.8 | 14.6 | 8.1 | 11.9 | 9.6 |
| $SO_2$ [µg m$^{-3}$] | 7.1 | 6.2 | 6.9 | 5.3 | 7.6 | 6.9 | 7.2 | 6.1 |
| GRad [W m$^{-2}$] | 310 | 276 | 338 | 346 | 225 | 240 | 113 | 122 |
| RH [%] | 40 | 45 | 41 | 44 | 54 | 61 | 70 | 78 |
| $T$ [°C] | 18.2 | 17.4 | 22 | 24 | 16.0 | 15.6 | 7.7 | 8.0 |
| WS [m s$^{-1}$] | 2.6 | 3.5 | 2.2 | 2.9 | 2.2 | 3.7 | 2.1 | 3.0 |
| $n$ [okta] | 2.6 | 2.5 | 4.4 | 4.8 | 5.6 | 5.3 | 6.4 | 6.3 |
| PBL [km] | 0.96 | 0.95 | 1.31 | 1.31 | 0.78 | 0.81 | 0.57 | 0.53 |
| $O_3$ [µg m$^{-3}$] | 76 | 83 | 93 | 96 | 61 | 75 | 46 | 52 |
| $O_3$ for prev. day [µg m$^{-3}$] | n.r. | 76 | n.r. | 62 | n.r. | 75 | n.r. | 49 |
| $N_{10-100}\times10^{-3}$ [cm$^{-3}$] | 8.5 | 8.3 | 8.2 | 2.6 | 6.5 | 7.3 | 4.1 | 2.6 |
| $PM_{10}$ mass [µg m$^{-3}$] | 21 | 21 | 20 | 16.9 | 26 | 22 | 28 | 23 |

n.r.: not relevant



**Table 5.** Range of $J_6$, GR and starting time $t_1$ in UTC+1 together with their mean values and standard deviations calculated jointly for the single events and onset 1 of the NPF events with double start (regional events), and separately for the onset 2 of the events with double start (urban events) in the near-city background of Budapest.

| Property | $J_6$ [cm$^{-3}$ s$^{-1}$] | GR [nm h$^{-1}$] | $t_1$ [hh:mm] |
|---|---|---|---|
| **Single + Onset 1 (43 days)** | | | |
| Range | 0.35–8.8 | 3.0–10.7 | 6:07–12:07 |
| Mean±st. deviation | 2.1±1.5 | 5.1±1.5 | 8:44±1:10 |
| **Onset 2 (8 days)** | | | |
| Range | 0.96–5.9 | 8.1–12.9 | 9:05–12:18 |
| Mean±st. deviation | 3.7±1.6 | 10.1±1.7 | 10:27±1:05 |




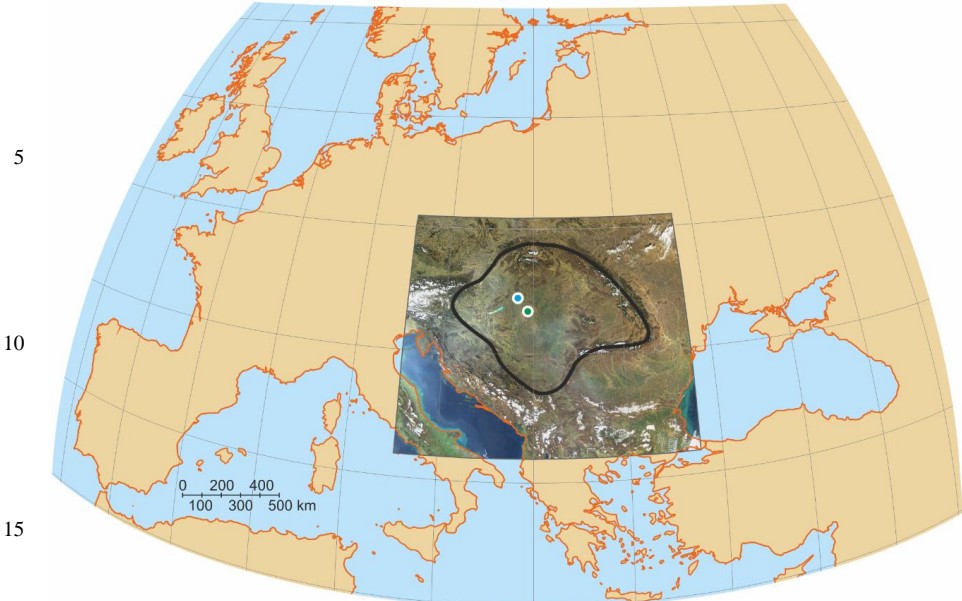

**Figure 1.** Image of the Carpathian Basin in Central Europe retrieved from Aqua/MODIS data indicating the location and advantages of a well separated basin (marked by black curve), and of the measurement sites in Budapest and at K-puszta station (shown by blue and green dots, respectively).



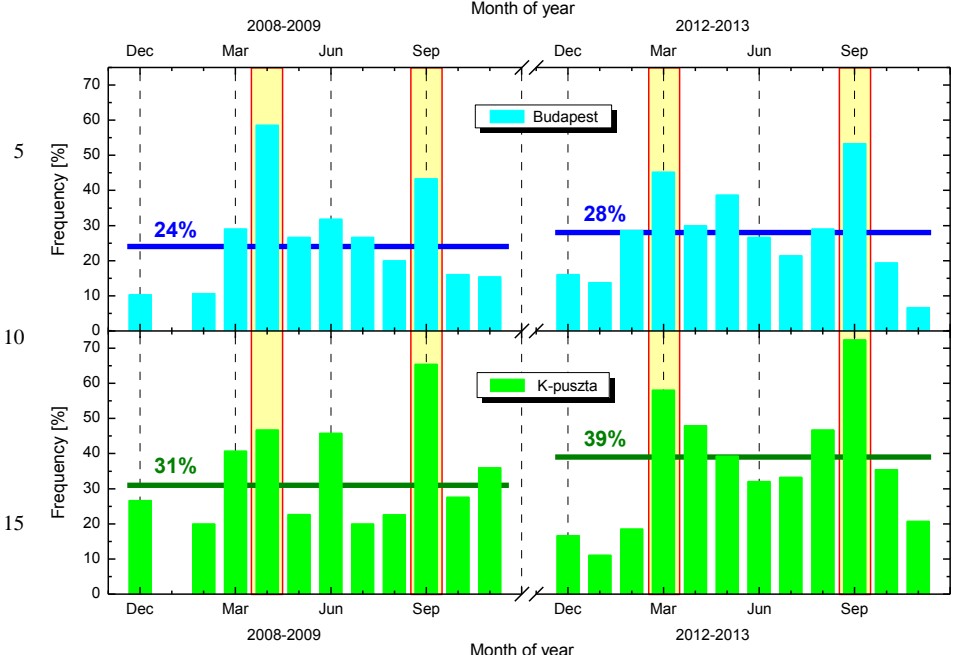

**Figure 2.** Distribution of monthly mean NPF frequency in Budapest and at K-puszta station. Annual mean frequencies are indicated by horizontal lines and figures. The spring and autumn maxima are highlighted in yellow bands.



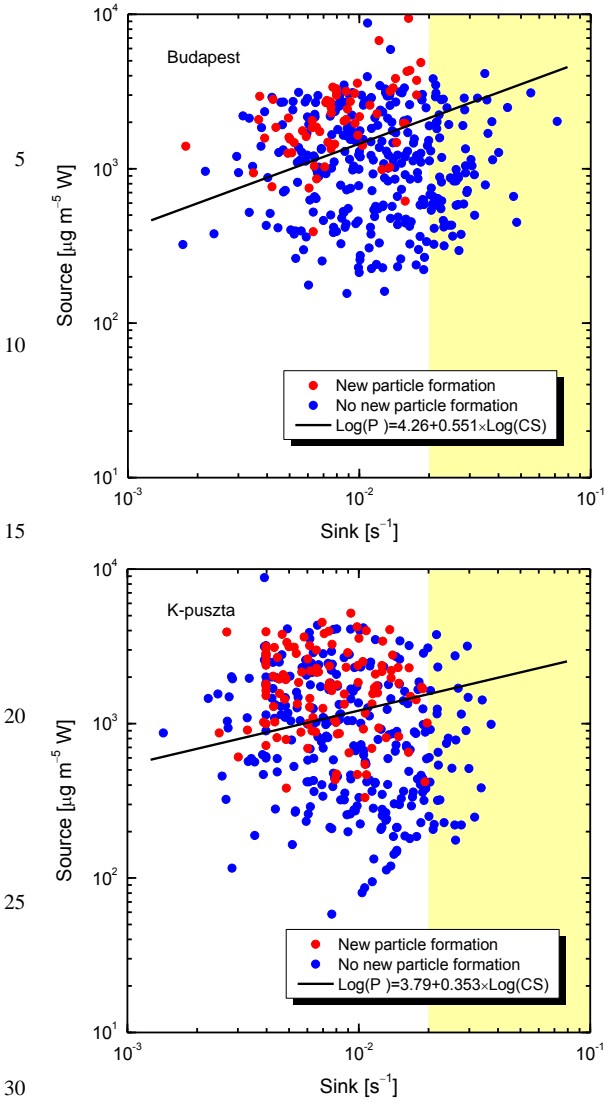

**Figure 3.** Relationship of the major source and sink of gas-phase $H_2SO_4$. The data are shown separately for the quantifiable-event days and non-event days in Budapest and at K-puszta station for the whole 2-year long time interval. Product $P$ of $SO_2$ concentration and global radiation was considered as the major source, and CS as the major sink. The yellow areas indicate the CS range in which the NPF events were supressed over the whole region.





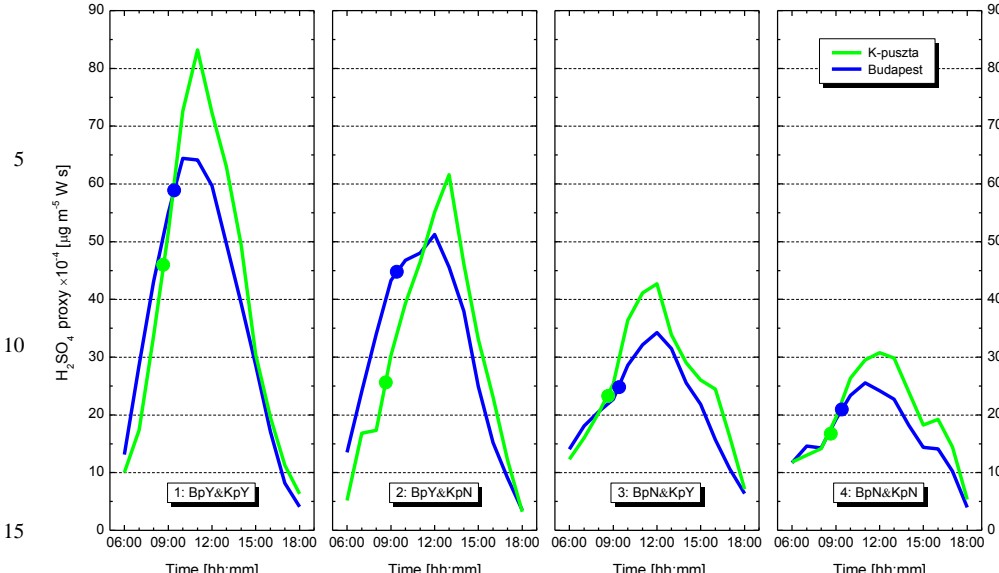

**Figure 4.** Hourly mean values of gas-phase $H_2SO_4$ proxy in Budapest and at K-puszta station averaged for the days when NPF events were identified in both Budapest and K-puszta station (BpY&KpY), event in Budapest and no event at K-puszta station (BpY&KpN), no event in Budapest and event at K-puszta station (BpN&KpY), and no event in both Budapest and K-puszta station (BpN&KpN). The proxy values at the mean starting time $t_1$ of the NPF are indicated by dots on the curves.




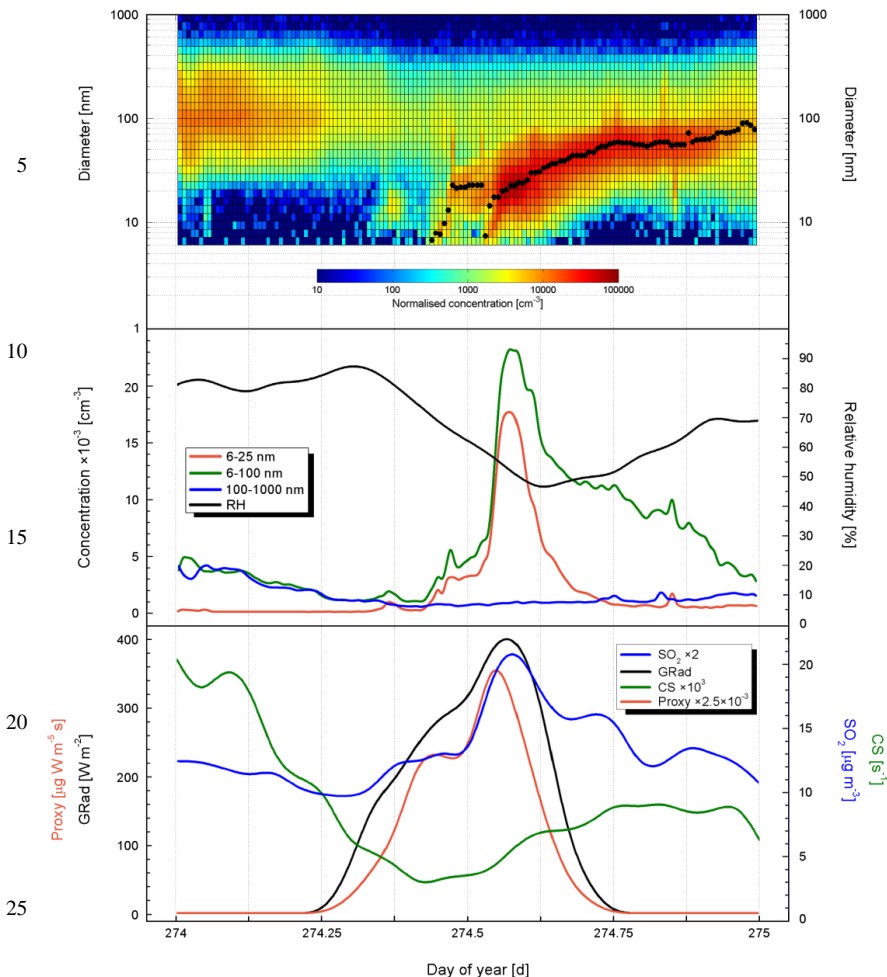

**Figure 5.** Surface plot (upper panel) for Sunday, 30 September 2012 showing a NPF and growth event with double consecutive start in the near-city background of Budapest. Time series of the fitted particle number median diameter for the nucleation mode are indicated by black dots. For the onsets 1 and 2, the $J_6$ were 1.34 and 5.9 cm$^{-3}$ s$^{-1}$, respectively, the GR were 7.7 and 12.9 nm h$^{-1}$, respectively, and the starting time $t_1$ were 9:53 and 12:18 current local time (UTC+2), respectively. Temporal evolution of particle number concentrations $N_{6-25}$, $N_{6-100}$ and $N_{100-1000}$, and of RH are displayed in the middle panel. The lower panel shows the diurnal variation of SO$_2$ concentration, GRad, CS and gas-phase H$_2$SO$_4$ proxy.





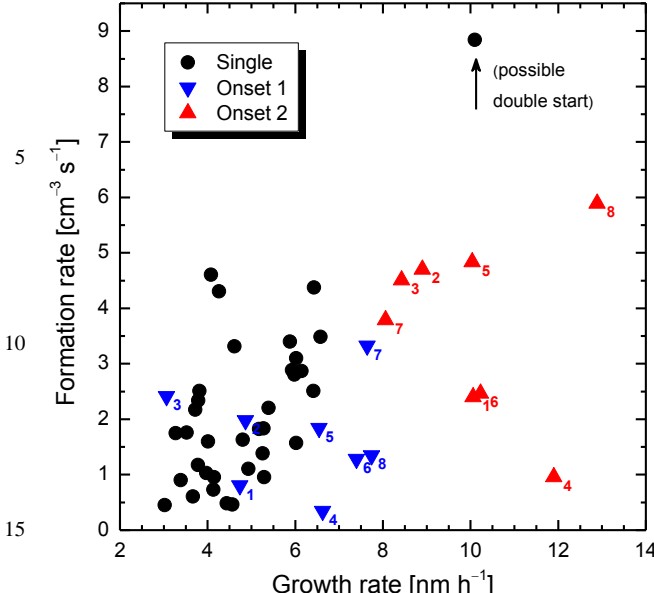

**Figure 6.** Dynamic properties for the NPF with single start, and for the onsets 1 and 2 of the events with double start in the
near-city background of Budapest. The data pairs corresponding to the two onsets of an event are indicated by numbers. The
data point marked as possible double onset (28 April 2012) was assigned to a weak onset 1 and a rather intensive onset 2 by
visual inspection although the identification of the two nucleation modes in the size distributions by fitting was not achieved,
likely due to the fluctuating data.





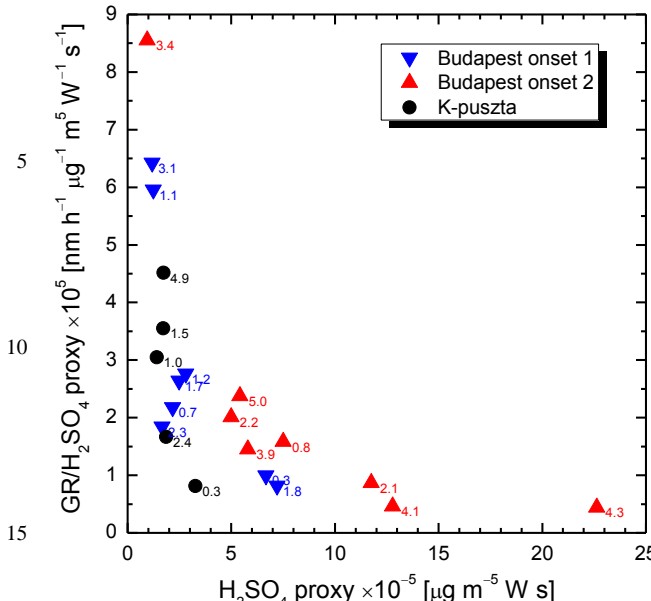

**Figure 7.** Dependence of the growth rate of NPF for a unity $H_2SO_4$ proxy on the proxy value separately for the onsets 1 and 2 of the events with double start in the near-city background of Budapest and for the regional NPF at K-puszta station on identical days. The numbers next to the symbols express the formation rate.