# Peer review of "Regional effect on urban atmospheric nucleation"

_Atmospheric Chemistry and Physics, 2016_

## Referee Comment (RC1) · Anonymous Referee #3 · 22 Apr 2016

Particle nucleation continues to be a major topic of research. Although nucleation in polluted atmospheres may play a less important role in atmospheric and climate science than it plays in the remote atmosphere, it can still have an important influence on particle size distributions and a major impact upon airborne concentrations of nanoparticles. This paper makes an interesting contribution by a careful analysis of nucleation events at two sites, one an urban site in Budapest and the other a rural site, K-puszta. The two sites are often connected by the wind trajectories, and the characteristics and relationship between nucleation events at the two sites are carefully intercompared. The work therefore allows an identification of conditions which favour nucleation at both sites simultaneously or at one site whilst not at the other. It therefore provides a useful contribution to the understanding of the determinants of the particle nucleation in a moderately polluted atmosphere. In this context it is worth noting that concentrations of sulphur dioxide and particulate matter in the measurement domain lie between those expected of remote rural locations and those occurring in the more polluted cities

of the world.

The data analysis is thorough and the work appears to be conceptually sound. There is, however, a lack of adequate information on the measurements of atmospheric criteria pollutants which are used heavily in the data analysis. It is indicated that these came from the closest measurement stations to the sites at which the nucleation studies were conducted, but further information is needed on the relative locations of the air quality network stations, and if possible, evidence on the local sources are spatial variability of air pollutant concentrations. Since these data come from a National Air Quality Network, it is assumed that quality assurance processes are appropriate, but a reference to relevant documentation or its inclusion in supplementary information would be reassuring.

One of the hypotheses proposed is that some nucleation evens at the K-puszta station were the result of oxidation of sulphur dioxide by stabilised Criegee intermediates, but the only evidence provided for this is an indication of increased ozone concentrations overnight before the nucleation events. As noted elsewhere in the paper, the higher ozone levels may be an indication of greater photochemical activity and could be associated with higher concentrations of hydroxyl radical. The formation of Criegee intermediates is dependent upon the oxidation of an alkene by ozone and no data are presented on the concentrations of alkenes. This process is invoked by the authors to explain some nucleation events at the K-puszta station but they do not consider the likely enhancement in anthropogenic alkenes at the Budapest site which offers a potential for formation of Criegie intermediates at that site also. Ozone concentrations measured at Budapest and K-puszta shown in Table 4 do not vary greatly and differences are smaller than in many urban/rural comparisons.

In addition to consideration of these points, there are two further recommendations. Firstly, the empirical relationship between the scaling factor k and GRad on page 3, line 34-35, requires units for GRad. Secondly, both the abstract (page 1, line 10) and the conclusions (page 9, lines 2-3) refer to the health risk associated with nanoparticle exposure. It is recommended that these references to health risk are removed. The body of evidence for health risks associated with airborne nanoparticle exposure remains relatively small and is not entirely coherent. It is also based very largely on urban environments dominated by traffic-generated nanoparticles and there is to date no evidence that the findings of these studies can necessarily be extrapolated to apply to nanoparticles deriving from atmospheric nucleation processes. Hence, the health impacts of particles nucleated in the European atmosphere remain a matter of conjecture.

---

## Referee Comment (RC2) · Anonymous Referee #1 · 26 Apr 2016

The manuscript presents a thorough analysis of observed new particle formation (NPF) and factors affecting the phenomenon in both rural and urban sites in the Carpathian basin. The authors discuss the similarities and differences between the two locations, concluding that the NPF is a common process that occurs over wide areas while there are local alterations in the features of the observed NPF events. The manuscript is well written and the results clearly presented. I recommend it for publication in ACP after the authors have addressed the following comments:

Specific comments:

1) Determining the quantities used in the analysis:
i) GR:
-How much does the choice of the method for assessing GR (the log-normal fitting

method) affect the obtained values? The maximum concentration method (see e.g. Kulmala et al., 2012) is also very often used; would the values change if this method was used instead?

-Is GR corrected for coagulational losses, as proposed by Leppä et al. (Atmos. Chem. Phys. 11, 4939–4955, 2011)?

-Are the GRs from literature (page 5, lines 39-41) determined for the same size range (and with the same method)?

ii) $J$:

-It is not clear how $J$ is in practice determined. The first paragraph of Section 2.3 lists the works by Kulmala et al. (2012), Kulmala et al. (2001) and Dal Maso et al. (2002) as references to how the DMPS data analysis was done; however, all these give somewhat different approaches for determining a formation rate $J$.

-Does $J$ depend on the determined GR as in the expression of Kulmala et al. (2012)? If so, the dependence should be brought up in the discussion for clarity.

-Is $J$ determined for $d_p$ = 6 nm for all the data? Information on the size is missing in e.g. Section 2.3, and Figs. 6 and 7. If the size is always 6 nm, how is $J$ determined for the K-puszta site, for which the lower limit of the DMPS is 10 nm?

iii) $t_1$:

It is stated that "The time $t_1$ also includes the time shift that accounts for the particle growth from the stable neutral cluster mode at (1.5+-0.4) nm to the smallest detectable diameter limit of the DMPS systems". How was this done exactly? Presumably this calculation requires a growth rate for the sub-detection sizes; which values were assumed? How much does the inclusion of this time shift affect the determined times $t_1$?

2) Page 4, lines 23-26: The quantity $\tau$ is used to assess if the air mass was transported from one site to the other for $\tau = 1$ and $\tau \ll 1$; for $\tau > 1$, it is only stated that "$\tau > 1$ is often caused by large (>7 m s–1) WSs." Can anything be hypothesised about the origin of the air mass in the last case? Also, $\tau$ isn't really discussed in the Results section; could it be e.g. added to Table 2?

3) Figure 3 and discussion on page 6: What does the dividing line describe, i.e. is there a physical reason to fit a line to the (sink, source)-data? (Isn't it quite clear also without the line that most of the red dots corresponding to events are at higher source values?)

4) Page 7, lines 21-22: Discussion on the effect of the condensation sink on NPF events: "This implies that the CS affected the NPF in the Budapest area, and that it can have preventing influence on the events. In contrast, the mean CS values for K-puszta station showed much less or even little effect." Why is the effect smaller in the K-puszta site? Are the absolute values of CS lower than in Budapest?

5) Page 8, line 23: "Fig. 8" should read "Fig. 7". Moreover, this Figure is only briefly mentioned in one sentence; it should be discussed more in the text to justify its existence.

6) Figures:
i) Figs. 4, 5 and 7: It would be useful for the reader to estimate the H2SO4 proxy also in units molec./cm$^3$ (e.g. as an additional y-axis), as well as the source in units molec./cm$^3$/s (Fig. 3).
ii) It would be useful to have also the hours, not only the day of year, in the time axis of Fig. 5. Also, in the bottom panel, the scaling factors could be written in the y-axes labels for clarity, and the legend could be removed (as now both the y-labels and the legend give essentially the same information)?

Technical comments:

-Page 1: The sentence "Despite the fact that NPF..." starting on line 28 should be tied to the previous sentence; on its own it doesn't mean anything. Also, the following sentence should be somehow modified, as it's not entirely clear to what kind of studies the expression "for such studies" refers.

-Page 5, lines 26-27: The English of the sentence "At present knowledge, advection of nucleating air masses cannot be excluded only in a few cases" is somewhat unclear; please modify.

-Page 7, lines 17-18: Tie the sentence "In spite of the fact that the estimated reduction..." to the previous sentence.

-Reference list: Seven references to works of one of the authors seems a bit unbalanced; the authors should cite also works other than their own in these occurrences.

---

## Referee Comment (RC3) · Anonymous Referee #2 · 3 May 2016

This is a very thorough study on the new particle formation (NPF) process taking place in a large but limited basin area of Hungarian puszta. The authors have carried out fine particle measurements along with gas and meteorological data, and when evaluating the results, they have scanned through many relevant parameters that are considered to affect NPF. The regional scales with local characteristics are considered. It seems to be that in many cases both sites are within the same NPF process.

There is also indication of distinct differences between urban (Budapest) and rural (K-puszta) sites: Budapest is urban , condensation sink is (on the average) higher than in rural, the NPFs start later (Fig 4) and they are also fewer than in rural. But, when the NPF does take place, then the values of both J6 and GR are systematically higher (Table 3), so it requires a higher threshold for everything to happen.

My critique here concerns why only the condensation sink is taken into consideration. What about all the other conventional air pollution parameters usually considered in

the atmospheric chemistry to be involved in the NPF. Like VOCs, NOx or ammonia? ELVOCS and mentioned but there is no data. There are some values such as O3 and SO2 given in Table 4 but I don't seem to get a clear complete picture on O3, SO2 from it. It is stated that SO2 does not count, or that NPF is not sensitive on SO2. It is finally concluded that CS and H2SO4 are the relevant parameters. And H2SO4 being a relevant parameter requires explaining part of the H2SO4 by introducing Criegee intermediate. But that, to my understanding, is not however explained by at least the O3 levels. The O3 levels seem to be, on the whole, apparently systematically higher in K-puszta. Please comment on this.

Also, according to Table 4, when there is NPF in Budapest but not in K-Puszta, still on the average/median the CS seems to be lower on K-puszta (6.8) than in Budapest (8.8). Please comment also on this.

To my opinion this is a good study, and the paper could be published, but prior to that the whole text within Chapters 3.2. and 3.3. should be clarified in what is actually claimed here.

---

## Author Comment (AC1) · 9 Jun 2016

The authors thank Referee #1 for his/her detailed and valuable comments to further improve and clarify the MS. We have considered all recommendations, and made the appropriate alterations. Our specific responses to the comments are as follows.

Comment 1.i Determining the quantities used in the analysis: GR: -How much does the choice of the method for assessing GR (the log-normal fitting method) affect the obtained values? The maximum concentration method (see e.g. Kulmala et al., 2012) is also very often used; would the values change if this method was used instead? -Is GR corrected for coagulational losses, as proposed by Leppä et al. (Atmos. Chem. Phys. 11, 4939–4955, 2011)? -Are the GRs from literature (page 5, lines 39-41) determined for the same size range (and with the same method)?

Response to Comment 1.i All GR values were determined by an identical method, i.e.

by the log-normal fitting method, and the possible influence of the different calculation methods was not studied. Uncertainties and possible systematic difference in dynamic properties caused by various evaluation approaches represent a relevant issue, which could be an objective of a separate dedicated study. The GR calculations were based on Kulmala et al., 2012, and the coagulation losses were not considered. The GR values cited from literature were not necessarily determined for the same size range and with the same method, and they serve for comparative purposes as the first orientation in the available data sets. It is also important to be aware of that there are banana plots with a broad onset of up to 3–4 hours, and the termination of their dynamic properties accurately represents a larger challenge than selection of the calculation method itself. In addition, it is worth mentioning here that an overview study on the global picture of observationally based estimate on NPF is under preparation by an international expert team, which is to handle the issue raised by the Referee.

Comment 1.ii J: -It is not clear how J is in practice determined. The first paragraph of Section 2.3 lists the works by Kulmala et al. (2012), Kulmala et al. (2001) and Dal Maso et al. (2002) as references to how the DMPS data analysis was done; however, all these give somewhat different approaches for determining a formation rate J. -Does J depend on the determined GR as in the expression of Kulmala et al. (2012)? If so, the dependence should be brought up in the discussion for clarity. -Is J determined for dp = 6 nm for all the data? Information on the size is missing in e.g. Section 2.3, and Figs. 6 and 7. If the size is always 6 nm, how is J determined for the K-puszta site, for which the lower limit of the DMPS is 10 nm?

Response to Comment 1.ii Formation rate J_d of particles with a diameter d nm was computed as:

J_d=dN_nuc/dt+CoagS_nuc x N_nuc + GR/delta(d) x N_nuc,

where N_nuc is the number concentration of nucleation-mode particles, CoagS_nuc is their coagulation scavenging efficiency, GR is the growth rate in the size range [d,

d+delta(d)], and t is time. The nucleated particles were estimated by N_6–25. It was assumed that the intensity of the NPF is constant for a certain time interval, and, hence, dN_6–25/dt was determined as the slope of the linear function N6–25 versus time within an interval where the dependence could be approximated by a linear fit. The referred publications indicate the continuous evolution of the calculation concept, and its final version (Kulmala et al., 2012) was adopted in the present MS. The text was modified now to express this accurately, and to emphasize also that J indeed depends on GR. Formation rates were determined for particles with a diameter of 6 nm, and it was specified or corrected at several places in the text and figures.

Comment 1.iii t1: It is stated that "The time t1 also includes the time shift that accounts for the particle growth from the stable neutral cluster mode at (1.5+-0.4) nm to the smallest detectable diameter limit of the DMPS systems". How was this done exactly? Presumably this calculation requires a growth rate for the sub-detection sizes; which values were assumed? How much does the inclusion of this time shift affect the determined times t1?

  Response to Comment 1.iii The time shift – that accounts for the particle growth from approximately 2 nm of the stable cluster mode (Kulmala et al., 2013) to the smallest detectable diameter limit of the DMPS systems (6 nm) – was calculated by adopting the GR value in the size window nearest to it in size space. This approximation can result in an underestimation of the shift by up to 30% since GR increases with d in this size range (Kulmala et al., 2012). It is noted that the shifts were mostly smaller than 30–40 min, which is acceptable with respect to the uncertainty of the starting time parameter t1, to the time resolution of the DMPS system, and to the ordinary dynamics of atmospheric processes. The text was extended to include these pieces of information. See the highlighted part of the marked-up MS.

Comment 2 Page 4, lines 23-26: The quantity tau is used to assess if the air mass was transported from one site to the other for tau = 1 and tau Âń 1; for tau > 1, it is only stated that " tau > 1 is often caused by large (>7 m s–1) WSs." Can anything be

hypothesised about the origin of the air mass in the last case? Also, tau isn't really discussed in the Results section; could it be e.g. added to Table 2?

Response to Comment 2 Quantity tau has sensible meaning for 1 column and 2 rows of Table 2, and therefore, it was not added there. Instead, we extended its discussion in the text. See the highlighted part of the marked-up MS. It is difficult to arrive at any solid conclusion for tau >1 since the related cases of very high WS happened only twice, and so, a representative evaluation could not be achieved.

Comment 3 Figure 3 and discussion on page 6: What does the dividing line describe, i.e. is there a physical reason to fit a line to the (sink, source)-data? (Isn't it quite clear also without the line that most of the red dots corresponding to events are at higher source values?)

Response to Comment 3 The dividing line in Fig. 3 was calculated by discriminant analysis. It was determined on one hand by the middle point between the arithmetic mean of the data subset for nucleation days and that for non-nucleation days, and on the other hand, it is perpendicular to the connecting interval between these 2 means (Hamed et al., 2010). We would like to keep the line in the figure since it is advantageous in fast and unambiguous orientation and serves as a discrete visual limit.

Comment 4 Page 7, lines 21-22: Discussion on the effect of the condensation sink on NPF events: "This implies that the CS affected the NPF in the Budapest area, and that it can have preventing influence on the events. In contrast, the mean CS values for Kpuszta station showed much less or even little effect." Why is the effect smaller in the K-puszta site? Are the absolute values of CS lower than in Budapest?

Response to Comment 4 Median values of CS for Budapest and K-puszta station are shown in Table 4 for four combinations of conditions, i.e for the time intervals when NPF events were identified in both Budapest and K-puszta station (BpY&KpY), event in Budapest and no event at K-puszta station (BpY&KpN), no event in Budapest and event at K-puszta station (BpN&KpY), and no event in both Budapest and K-puszta station

(BpN&KpN). It turns out from them that the CS varies in a broader range in Budapest among these cases and that its high values are associated with no NPF, while CS changes in a smaller interval at K-puszta station. The CS depends sensitively on the concentration and size-distribution of pre-existing aerosol particles. At K-puszta station, the average particle number concentrations are substantially smaller, and hence, the CS values and their changes are smaller as well. The text was extended to express these more precisely. See the highlighted part of the marked-up MS.

Comment 5 Page 8, line 23: "Fig. 8" should read "Fig. 7". Moreover, this Figure is only briefly mentioned in one sentence; it should be discussed more in the text to justify its existence.

Response to Comment 5 The figure number was corrected, and it was also more discussed now. See the highlighted part of the marked-up MS.

  Comment 6 Figures: i) Figs. 4, 5 and 7: It would be useful for the reader to estimate the H2SO4 proxy also in units molec./cm3 (e.g. as an additional y-axis), as well as the source in units molec./cm3/s (Fig. 3). ii) It would be useful to have also the hours, not only the day of year, in the time axis of Fig. 5. Also, in the bottom panel, the scaling factors could be written in the y-axes labels for clarity, and the legend could be removed (as now both the y-labels and the legend give essentially the same information)?

Response to Comment 6 The gas-phase H2SO4 proxy value was calculated as [SO2]$\times$GRad/CS for intensities >10 W m–2. Its average or extreme values for Budapest and K-puszta station were also expressed in absolute concentrations for several cases as well by using the scaling factor k between the proxy value and H2SO4 concentration of k=$1.4\times10$ˆ$-7\times$ GRadˆ$-0.70$ (Petäjä et al., Sulfuric acid and OH concentrations in a boreal forest site, Atmos. Chem. Phys. 9, 7435–7448, 2009). These concentrations were given in Table 4 or on page 6, lines 23–24. As far as the figures are concerned, we would prefer using the proxy without adopting the scaling factor

since it was derived specifically for a remote boreal site as an empirical relationship. Urban areas are expected to differ from remote regions (Mikkonen et al., A statistical proxy for sulphuric acid concentration, Atmos. Chem. Phys. 11, 11319–11334, 2011), and the GRad involved implicitly in the scaling factor can distort the relationships and trends investigated in the figures. We emphasize this by a separate note which was added now into the text, and with a new reference. See the highlighted part of the marked-up MS. In addition, we also think that Fig. 5 showing the size distribution surface plot and 8 related meteorological, pollutant gas an aerosol data in 3 panels would possibly become over-sophisticated to follow by an extra axis. The minor ticks of the time axis on Fig. 5 represent 3 hour time intervals, and thus, the hourly dependency for a day can be recognised. We originally selected the way of indicating the figure scaling factors for various independent variables in the figure legends. This seems virtually equivalent with the solution suggested by the Referee.

Technical comments: -Page 1: The sentence "Despite the fact that NPF..." starting on line 28 should be tied to the previous sentence; on its own it doesn't mean anything. Also, the following sentence should be somehow modified, as it's not entirely clear to what kind of studies the expression "for such studies" refers. -Page 5, lines 26-27: The English of the sentence "At present knowledge, advection of nucleating air masses cannot be excluded only in a few cases" is somewhat unclear; please modify. -Page 7, lines 17-18: Tie the sentence "In spite of the fact that the estimated reduction..." to the previous sentence. -Reference list: Seven references to works of one of the authors seems a bit unbalanced; the authors should cite also works other than their own in these occurrences.

Response to Technical comments The whole MS was checked for typing and language errors. The specific examples mentioned by the Referee were, naturally, all corrected. See the highlighted part of the marked-up MS. There are a few and recent publications on the NPF in the Carpathian Basin. They were cited since they were regarded to be relevant for the context and background of the special objective of the present paper.

Their number is, however, not extraordinary large in comparison with the total number of references.

---

## Author Comment (AC2) · 9 Jun 2016

The authors thank Referee #2 for his/her valuable comments to further improve and clarify the MS. We have considered all recommendations, and made the appropriate alterations. Our specific responses to the comments are as follows.

Comment 1 My critique here concerns why only the condensation sink is taken into consideration. What about all the other conventional air pollution parameters usually considered in the atmospheric chemistry to be involved in the NPF. Like VOCs, NOx or ammonia? ELVOCS and mentioned but there is no data. There are some values such as O3 and SO2 given in Table 4 but I don't seem to get a clear complete picture on O3, SO2 from it. It is stated that SO2 does not count, or that NPF is not sensitive on SO2.

Response to Comment 1 We were measuring several criteria air pollutant concentrations for the urban and rural sites over 2-year time interval. VOCs, NOx and NH3 were

unfortunately not involved because of several reasons. Continuous measurements of monoterpenes – as one of the most important groups of VOCs – are usually not available or they are scarce so far. Their proximity value has been just elaborated and introduced very-very recently. We plan to adopt it in the future work. The measured variables made it feasible to arrive at the conclusions on the urban-regional similarities and/or differences. The parts dealing with the role and importance of O3 and SO2 were reformulated to clarify our conclusions better. See the corresponding highlighted part of the marked-up MS.

Comment 2 It is finally concluded that CS and H2SO4 are the relevant parameters. And H2SO4 being a relevant parameter requires explaining part of the H2SO4 by introducing Criegee intermediate. But that, to my understanding, is not however explained by at least the O3 levels. The O3 levels seem to be, on the whole, apparently systematically higher in K-puszta. Please comment on this.

Response to Comment 2 The main oxidation process of SO2 by OH radical could not completely explain the NPF occurrence through the formation of H2SO4 in the forested site of K-puszta, and the missing contribution was related to the effect of stabilized Criegee intermediates (CIs) as the likely oxidising agent. Stabilized CIs are formed by ozonolysis of unsaturated organics including terpenoid compounds (Mauldin III et al., 2012). These are emitted in large amounts by plants. In this sense, O3 plays a mediation role in the process. The text was modified accordingly to express our intention more clearly. See the highlighted part of the marked-up MS.

Comment 3 Also, according to Table 4, when there is NPF in Budapest but not in K-Puszta, still on the average/median the CS seems to be lower on K-puszta (6.8) than in Budapest (8.8). Please comment also on this.

Response to Comment 3 The basic preconditions of NPF events are realised by competing source and sink terms, which can largely vary in different atmospheric environments. As a consequence, NPF can occur even in polluted environments (with

large condensation and scavenging sinks) if the sources for condensable chemical species are even larger, and the other conditions are also favourable (Salma et al., Measurement, growth types and shrinkage of newly formed aerosol particles at an urban research platform. Atmos. Chem. Phys., doi:10.5194/acp-2016-239, in production, 2016). Furthermore, it is worth mentioning here that NPF occurrence depends on a complex set of multiple variables at a time. All of them contain relevant information while it cannot be expected that any standalone property or paired relationship can explain or even directly be linked to the NPF occurrence. The text was extended to include this point as well. See the highlighted part of the marked-up MS.

Comment 4 To my opinion this is a good study, and the paper could be published, but prior to that the whole text within Chapters 3.2. and 3.3. should be clarified in what is actually claimed here.

Response to Comment 4 Sections 3.3 and 3.4 (according to the new numbering) were revised to emphasize our arguments and conclusions in a more explicit and precise way. See the highlighted parts of the marked-up MS.

---

## Author Comment (AC3) · 9 Jun 2016

The authors thank Referee #3 for his/her valuable comments to further improve and clarify the MS. We have considered all recommendations, and made the appropriate alterations. Our specific responses to the comments are as follows.

Comment 1 The data analysis is thorough and the work appears to be conceptually sound. There is, however, a lack of adequate information on the measurements of atmospheric criteria pollutants which are used heavily in the data analysis. It is indicated that these came from the closest measurement stations to the sites at which the nucleation studies were conducted, but further information is needed on the relative locations of the air quality network stations, and if possible, evidence on the local sources are spatial variability of air pollutant concentrations. Since these data come from a National Air Quality Network, it is assumed that quality assurance processes are appropriate, but a reference to relevant documentation or its inclusion in supplementary information

would be reassuring.

Response to Comment 1 The municipal air quality measurement stations perform regular measurements of criteria air pollutants at several locations in Budapest. More detailed information was added on the location, instrumentation of and measurements at the closest municipal stations. It was also noted now that SO2 concentration is ordinary distributed without larger spatial differences within the city (Salma et al., Comprehensive characterisation of atmospheric aerosols in Budapest, Hungary: physicochemical properties of inorganic species, Atmos. Environ. 35, 4367–4378, 2001), and, therefore, its actual value at the BpART research platform is less influenced by air mass directions. In addition, an important advantage of the selected urban location at the river Danube is that it receives well-mixed, averaged air masses from the city centre. The text was extended to include these pieces of information, and a reference was also added. See the highlighted part of the marked-up MS.

Comment 2 One of the hypotheses proposed is that some nucleation evens at the K-puszta station were the result of oxidation of sulphur dioxide by stabilised Criegee intermediates, but the only evidence provided for this is an indication of increased ozone concentrations overnight before the nucleation events. As noted elsewhere in the paper, the higher ozone levels may be an indication of greater photochemical activity and could be associated with higher concentrations of hydroxyl radical. The formation of Criegee intermediates is dependent upon the oxidation of an alkene by ozone and no data are presented on the concentrations of alkenes. This process is invoked by the authors to explain some nucleation events at the K-puszta station but they do not consider the likely enhancement in anthropogenic alkenes at the Budapest site which offers a potential for formation of Criegie intermediates at that site also. Ozone concentrations measured at Budapest and K-puszta shown in Table 4 do not vary greatly and differences are smaller than in many urban/rural comparisons.

Response to Comment 2 Ambient concentrations of isoprene and mono-terpenes were measured earlier with a high-frequency PTR-MS to be between 0.028–0.82 ppbv and

0.019–0.63 ppbv, respectively (Maenhaut et al., 2008). Similar data for Budapest are, unfortunately, missing. It is expected, however, that concentrations of VOCs, including alkenes, are considerably smaller in the city than at the forested site of the K-puszta station. We extended the text with this sensible assumption. See the highlighted part of the marked-up MS.

Comment 3 Firstly, the empirical relationship between the scaling factor k and GRad on page 3, line 34-35, requires units for GRad.

Response to Comment 3 We added the units for GRad now. See the highlighted part of the marked-up MS.

Comment 4 Secondly, both the abstract (page 1, line 10) and the conclusions (page 9, lines 2-3) refer to the health risk associated with nanoparticle exposure. It is recommended that these references to health risk are removed. The body of evidence for health risks associated with airborne nanoparticle exposure remains relatively small and is not entirely coherent. It is also based very largely on urban environments dominated by traffic-generated nanoparticles and there is to date no evidence that the findings of these studies can necessarily be extrapolated to apply to nanoparticles deriving from atmospheric nucleation processes. Hence, the health impacts of particles nucleated in the European atmosphere remain a matter of conjecture.

Response to Comment 4 We completely removed the first reference (page 1, line 10) to the human health. Considering the potential importance of the health effects of UF particles in cities, we would like to keep the second references as an outlook. We can accept the arguments of the Referee, and modified its formulation according to the Referee's requirement now as a working hypothesis to express that 1) the results for health effects for healthy adults obtained so far are not conclusive, 2) further dedicated studies are needed to access the health significance of NPF.